# Quantifying the Measurement Error on England and Wales EPC Ratings

**Jenny Crawley [1],\*, Phillip Biddulph [1], Paul J. Northrop [2], Jez Wingfield [1] 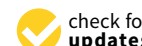, Tadj Oreszczyn [1] and Cliff Elwell [1]**

[1] UCL Energy Institute, University College London, London WC1H 0NN, UK; p.biddulph@ucl.ac.uk (P.B.); j.wingfield@ucl.ac.uk (J.W.); t.oreszczyn@ucl.ac.uk (T.O.); clifford.elwell@ucl.ac.uk (C.E.)

[2] Department of Statistical Science, University College London, London WC1H 0NN, UK; p.northrop@ucl.ac.uk

\* Correspondence: jenny.crawley@ucl.ac.uk; Tel.: +44-20-3108-5971

**Abstract:** Domestic Energy Performance Certificates (EPCs) are used in the UK to provide energy efficiency ratings for use in policy and investment decisions on individual dwellings and at a stock level. There is evidence that the process of creating an EPC introduces measurement error such that repeat assessments of the same property give different ratings, compromising their reliability. This study presents a novel error analysis to estimate the size of this effect, using repeated EPC assessments of 1.6 million existing dwellings in England and Wales. A statistical model of how measurement error contributes to variation between repeated measurements is set out, and exploratory data analysis is used to decide how to apply this model to the available data. The results predict that the one standard deviation measurement error decreases with EPC rating, from around ± 8.0 EPC points on a rating of 35 to ±2.4 on a rating of 85. This predicted error is higher than the limit recommended in UK guidance except in very efficient buildings; it can also result in dwellings being rated in the wrong EPC band, for example it was estimated that 24% of band D homes are rated as band C.

**Keywords:** uncertainty; energy performance certificates; domestic buildings; dataset; mathematical modelling

## 1. Introduction

Domestic Energy Performance Certificates (EPCs) are short reports providing an estimate of the energy and environmental performance of a dwelling operating under a set of normative technical and operational assumptions. EPCs were introduced by the European Union in 2003 with the aim of encouraging energy efficiency in buildings through the provision of information to owners and tenants, firstly by allowing them 'to compare and assess [a building's] energy performance' and secondly to give 'recommendations for the cost-optimal or cost-effective improvement of the energy performance of a building or building unit' [1]. This article focusses on EPCs in England and Wales, which have been mandatory for selling or letting out dwellings since 2008.

EU member states may produce EPCs using either calculated or measured energy consumption [1]; England and Wales adopted the former methodology. The input data for the calculation is provided through an assessment of building and heating system characteristics by a qualified assessor, using both on-site inspection and documentation. The assessment documents the fabric and heating system present, but not its condition. The data is then entered into an energy model using a standardised heating regime to calculate the energy performance of a building under standard occupancy and weather. In the UK, the assessment of new dwellings is carried out using an energy model called the

Standard Assessment Procedure (SAP), which predates EPCs [2,3]; existing dwellings use a version with reduced input data and heavier use of default assumptions called Reduced Data SAP (RdSAP). One output from this process is Energy Efficiency Rating or SAP rating [4], a function of modelled fuel cost which is either linear or logarithmic depending on the magnitude of the fuel cost. This results in an integer value between 1 (lowest) and 100 (highest except in cases of net exporting buildings) which in turn can fall into one of seven energy efficiency bands from G (lowest) to A (highest).

In this study, the numeric score will be referred to as 'EPC rating' and the band it falls into as 'EPC band'. The mapping between ratings and bands is shown in Table 1.

**Table 1.** EPC ratings mapped to EPC bands.

| EPC band | G | F | E | D | C | B | A |
|----------|------|-------|-------|-------|-------|-------|-----|
| EPC rating | 1–20 | 21–38 | 39–54 | 55–68 | 69–80 | 81–91 | 92+ |

There is mixed evidence as to whether EPC ratings have yielded the outcomes aspired to by the EU of improved householder awareness of energy efficiency and increased motivation to upgrade properties [4,5]. However, they have been adopted by the UK government as a metric to inform decisions and policies. For example, the Feed-in Tariff subsidy for renewable energy generation requires a dwelling to achieve an EPC band D or above to gain a higher tariff rate [6], and under Minimum Energy Efficiency standards introduced in 2018 dwellings rated below band E cannot be let out [7].

There is extensive research documenting discrepancies between predictions made on EPC certificates and in-use energy performance [2,8,9], even though predicting operational energy consumption is not the stated aim of EPCs or their underlying models. Less research focuses on internal variation in the process used to create EPC ratings, caused by a combination of assessor variation and changes to the algorithms used to calculate the ratings.

The UK government has made the entire dataset of English and Welsh domestic EPC ratings from 2008 to 2016 publicly available, consisting of around 15 million records. In this study, this dataset is used to estimate the uncertainty in EPC rating on individual certificates.

## 2. Background

### 2.1. Previous Work on Uncertainty on EPC Certificates

A domestic EPC assessment is a survey carried out of a dwelling by a qualified expert [1], recording features of the building and energy systems which are required for input into the energy models. The potential of this type of assessment and data input process to introduce error into the subsequent calculation was highlighted decades before the introduction of EPCs, with Chapman's 1991 article discussing how increasing data requirements should decrease the model error but increase the input error [10]. Since the introduction of EPCs in the EU, there has been awareness that input data error leads to uncertainties in calculated energy consumption [11,12]; more recently, the uncertainties in EPC rating using the alternative approach—measured energy data—have also been discussed for the case of Sweden [13]. The question then becomes: how large an uncertainty is acceptable for an EPC to still be useful? There is no EU-level requirement on this; the Buildings Performance Institute Europe (BPIE) comments on this (without stating a justification):

> "For instance, in the context of label classes, a deviation of one label class is generally acceptable while a deviation of two or more label classes may undermine the credibility of the certificate and hence may be regarded as unacceptable." [14]

EU member states must develop quality assurance guidance for the EPC creation process [1]. The UK's implementation of this consists of regular audits to check assessors against a target that

'95% of a random sample of EPCs are within + or −5 SAP points of the "truth"' [15]. In this context, the 'truth' is determined by the quality assurance assessor.

In the UK, the reliability of EPC certificates attracted heightened interest around 2012 with the launch of the 'Green Deal' policy, a scheme for financing energy efficiency measures through energy bill savings, which relied on the validity of the EPC calculation on a per-dwelling basis [16]. A study carried out by Jenkins et al. [17] investigated the variation in EPC ratings in 29 dwellings, each undergoing four assessments using a 'mystery shopper' approach, complemented by one reference assessment. The average difference between the highest and lowest EPC rating for the same property was 11.1 EPC points, with ratings in almost two-thirds of the 29 dwellings varying by at least two EPC bands. The authors found indicative evidence of wider variation in older properties, but the sample size was not large enough for this to be statistically confirmed.

Recently, Hardy and Glew [18] conducted a large-scale study flagging identifiable errors in the England and Wales dataset (introduced below), such as repeat assessments in which built form variables differed or energy efficiency measures were apparently removed. The authors predicted that 36–62% of certificates contained at least one such identifiable error, and that where one or multiple errors were discovered and corrected by assessors and a new certificate lodged, this led to a change in rating of 4 EPC points.

### 2.2. The England and Wales EPC Dataset

The dataset used in this article is the first release of the public dataset of EPC ratings, containing around 15 million records from 13 million dwellings, collected from 2008 to 2016 and reported with some limited information on the inputs to the underlying SAP/RdSAP calculation. The majority of the dataset comprises of existing dwellings (91%, 12 million dwellings). The remaining records, associated with new dwellings, are excluded from this analysis due to systematic differences between their calculation methodology—SAP, and the version used for existing dwellings—RdSAP. However, even within the sample of existing dwellings with EPC certificates, the RdSAP calculation methodology has undergone several changes over the timescale of the data collection. These changes do not appear to significantly affect the overall distribution of EPC ratings over time, as shown in Figure 1. Nevertheless, calculation methodology change over time is compounded with other sources of variation, such as differences in the software implementation of the methodology between approved EPC schemes and differences between assessors (the focus of Jenkins et al. [17]), to create a total uncertainty on EPC rating: the focus of the current study.

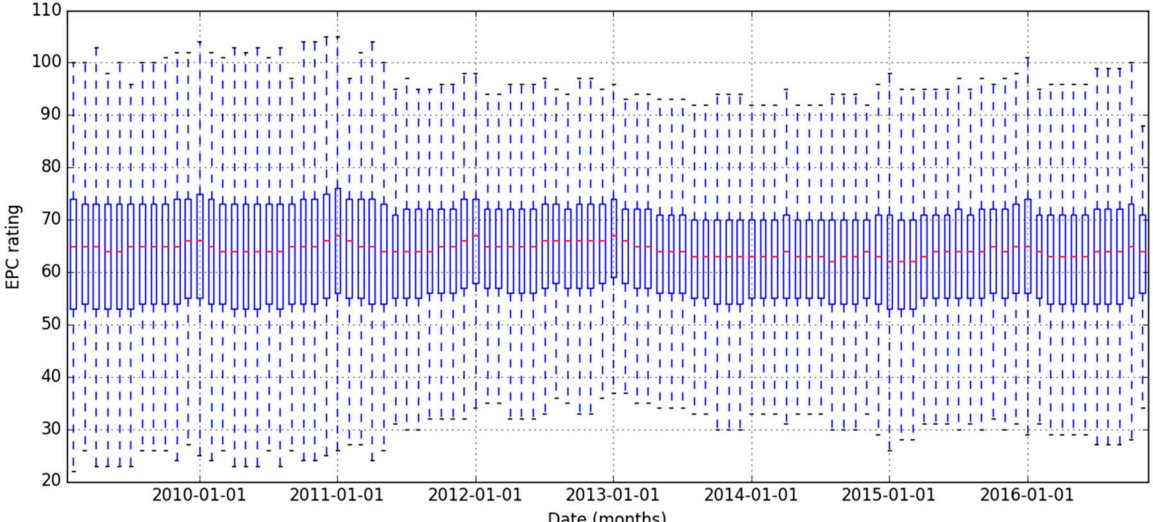

**Figure 1.** Monthly distributions of EPC ratings from 2009 to 2016. Showing median (red), interquartile range (edges of each box) and 1.5 interquartile ranges above and below the box (edges of whiskers).

Within this dataset, a subset of 1.6 million existing dwellings had exactly two EPCs recorded. The distribution of first and second ratings for these dwellings is shown in Figure 2. The distributions are fairly similar, although the second ratings are slightly more narrowly distributed and peak slightly higher than the first ratings. Both have an unusual peak at 1; this will be returned to in Section 5.

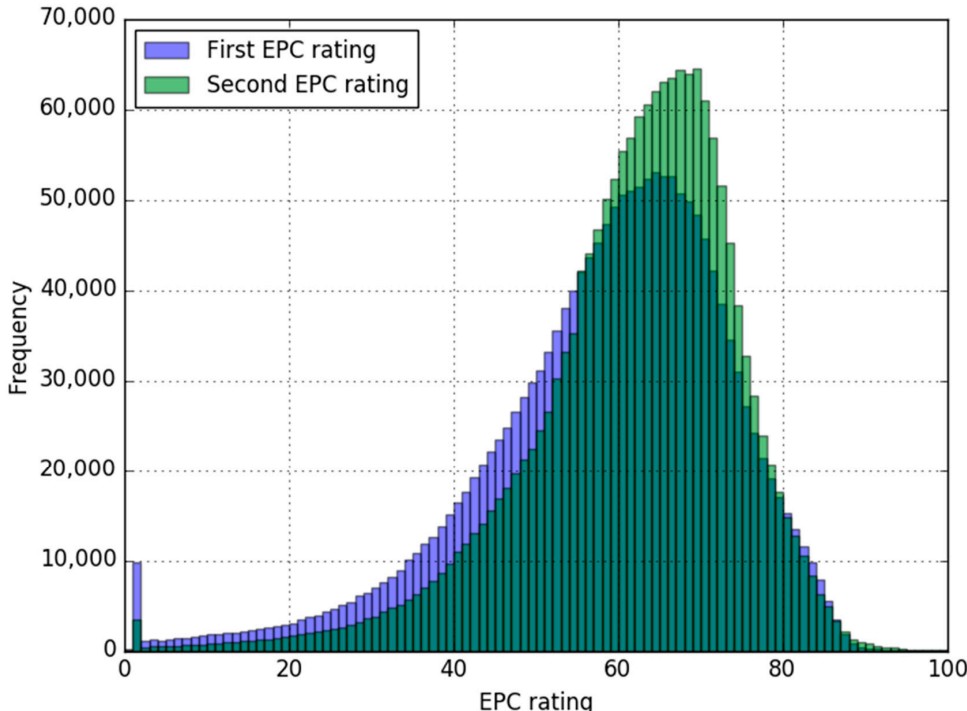

**Figure 2.** Histograms of first and second EPC ratings for the subset in the England and Wales dataset which have two ratings.

The valid lifetime of an EPC is 10 years, longer than the timespan of the dataset used in this article; therefore, the expiry and replacement of certificates should not explain the presence of multiple certificates per home. Possible reasons why two assessments may have been undertaken between 2008 and 2016 are summarised below; note that this information is not recorded in the dataset except that identified in the first bullet point.

- Several renewable or energy efficiency subsidies require a recent or up-to-date EPC (e.g., Ref. [6]); others require a post-installation EPC (e.g., Ref. [19]).
- A mistake discovered in an assessment or lodgement cannot be overwritten, and a new corrected EPC must therefore be lodged [20]. However, the older, defective EPC may remain in the database in cases where the assessor neglects to cancel it.
- Some property owners may not realise an EPC already exists for their dwelling.
- Providing an EPC to new tenants could bring extra revenue to estate agents.
- Landlords or homeowners whose properties have undergone significant changes may wish to obtain a new EPC, particularly where those changes may improve saleability or rentability.
- Social landlords with large portfolios of dwellings will carry out regular stock assessments and associated improvement works which will sometimes require EPCs on a more regular basis.

It may be expected that in the absence of assessor variation and significant changes in calculation methodology, if energy-related changes have been made to the fabric or building services, then the second EPC rating would in most cases be higher than the first. If no such changes have been made, then the second EPC rating should be equal to the first. Therefore, EPC rating should not decrease over time. This assumption was also used by Filippidou et al. [21] in their analysis of Dutch retrofits;

properties whose EPC rating decreased after retrofit were excluded from the analysis on the basis of error.

## 2.3. Observations of Movement between Subsequent Ratings

In the England and Wales dataset, it is very common for EPC ratings to decrease as well as increase. In this section, the dataset of repeated measurements is examined in terms of its per-dwelling movement, as opposed to the group-level histograms in Figure 2, to illustrate this point.

A visualisation of the movement from first EPC band to second EPC band is shown in Figure 3. The lines on the diagram represent transitions from each band to each band, and their width indicates the number of dwellings undergoing each particular transition (transitions containing less than 1% of dwellings are not shown). Some examples are now given from the underlying data used to make this visualisation. Dwellings rated as band D are most likely to remain in this category, but 23% move up to C, and 15% move down to band E. Two percent move two bands up or down, to bands B and F. Furthermore, dwellings initially rated as A can move as far down as band F.

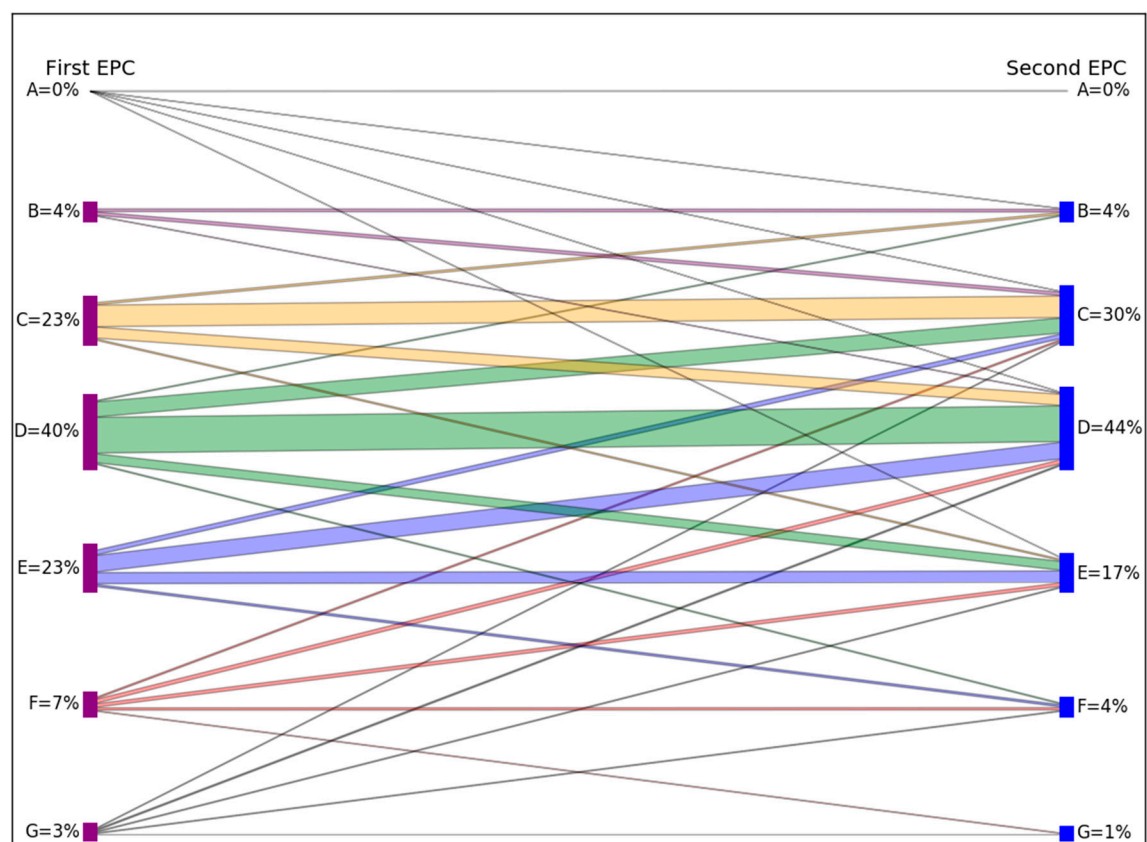

**Figure 3.** Visualisation of change in EPC band from first to second assessment, at a dwelling level. Line thickness represents number of dwellings making each particular transition. Transitions with less than 1% of dwellings are excluded for clarity.

Summing the data in Figure 3 yields that 48% of dwellings stay in the same band upon second measurement, whilst 33% move to a higher band and 19% to a lower band. Whilst there is significant movement in both directions, this also highlights an asymmetry as more increase than decrease, as seen previously in Figure 2.

An alternative visualisation of the data is shown in Figure 4, this time using numeric EPC ratings instead of categorical EPC bands. Here, the spread around the diagonal line indicates movement between ratings, but the spread is not exactly symmetric around the diagonal.

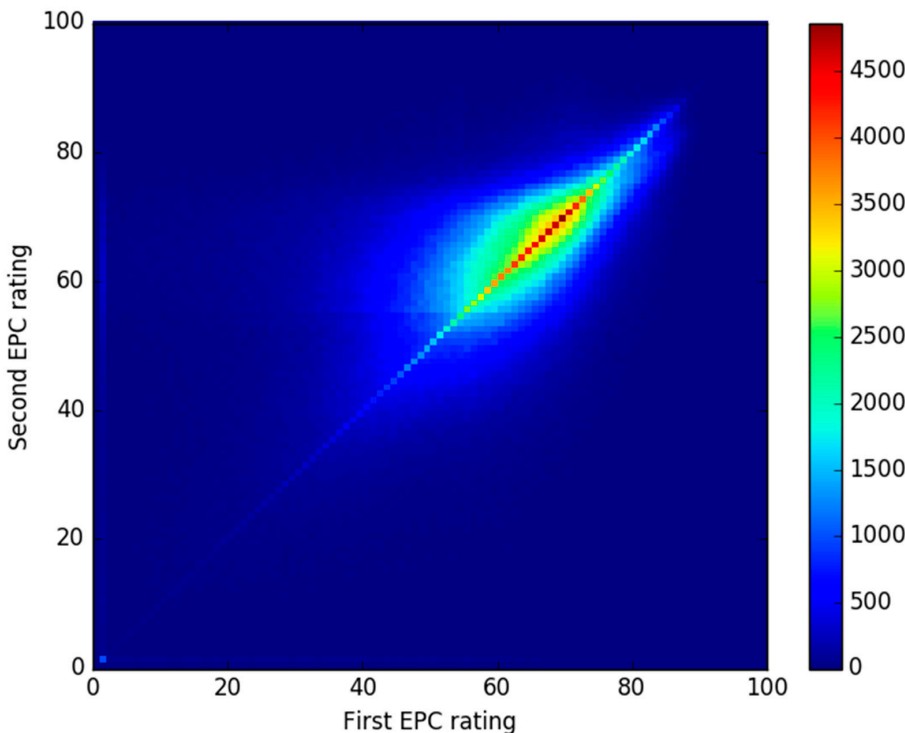

**Figure 4.** Density plot of first and second EPC ratings for 1.6 million dwellings with two ratings in the England and Wales dataset. Colour indicates number of dwellings.

### 2.4. Research Question and Key Aim of Study

Observation of the changes in EPC rating between subsequent assessments allows a theory to be generated that a proportion of this movement is explained by what shall be here-termed *measurement error*, arising from assessor, RdSAP version and software implementation variation. This in turn leads to a policy relevant question: what is the uncertainty on EPC ratings caused by this measurement error?

This question is not answerable using the current dataset, since it is not possible to prove the causal link between EPC rating movement and the possible sources of variation described above. Therefore, the above question is reframed below to form the research question to be addressed in this study:

*What level of measurement error is consistent with the observed data?*

The key aim of the study is to use the observed difference between two EPC ratings for the same dwelling, across a very large sample of dwellings, to characterise and estimate the measurement error on the rating process.

Before describing the methodology used, it is necessary to set out the mathematical link between divergence in observations from repeated measurements and measurement error. This theory can be used later to guide the data analysis.

## 3. Theory

### 3.1. Measurement Error

An EPC rating can be conceptualised as a measurement. In reality, it is not one single measurement but the outcome of a calculation involving many individual measurements and assumptions. However, for the purposes of explaining the variation in ratings of the same property, the rating itself will be considered as a measurement.

The theory of making a physical measurement relates a measured value *M* to a 'true value' *T* via Equation (1) [22]:

$$M = T + e \tag{1}$$

e represents the measurement error, and in many disciplines is usually assumed to be drawn from a Gaussian distribution with a standard deviation $\sigma$. Under this assumption, the interpretation of $\sigma$ has a physical meaning as follows: in 68% of measurements, the measured value obtained would lie within $\pm\sigma$ of the true value.

In the context of EPC ratings, Equation (1) can be expressed as:

$$EPC_{measured} = EPC_{true} + \Delta \tag{2}$$

$EPC_{measured}$ refers to the rating obtained by the assessor. $EPC_{true}$ is the true value which would be obtained with a perfect assessor and unchanging assessment methodology, and $\Delta$ is the measurement error.

There are several unknowns regarding the EPC measurement error $\Delta$. Firstly, it is not known whether a Gaussian shape is the true form of the error on EPCs. Secondly, if $\Delta$ is drawn from a Gaussian distribution with standard deviation $\sigma$, then it is not known how $\sigma$ might vary across different dwelling types or other contextual variables.

This research will explore where it is appropriate to use the conceptualisation of measurement error as the standard deviation of a Gaussian distribution. Where this is possible, the research question can be answered by estimating $\sigma$ in different regions of the data.

*3.2. Using Repeated Observations to Estimate Measurement Error*

The only known term from Equation (2) is $EPC_{measured}$, however, two data points for this term are available for each dwelling. These measured values will be termed EPC1 and EPC2 for the first and second measurement in time respectively, with a per-dwelling mean termed <EPC> as in Equation (2).

$$< EPC > = var\left(\frac{EPC1 + EPC2}{2}\right) \tag{3}$$

If a dwelling has not been modified between assessments, <EPC> is the best estimate of $EPC_{true}$. However, if a dwelling has undergone refurbishments between assessments, there is no longer a single value for $EPC_{true}$ and therefore <EPC> is no longer the best estimate for $EPC_{true}$.

The distance between <EPC> and either of the two data points per dwelling is a metric describing how different two measurements of the same dwelling are. A shorthand for this metric will be defined here as Y:

$$Y = < EPC > - EPC1 \tag{4}$$

Note that EPC2 could have been used instead of EPC1 in Equation (4), giving Y the same magnitude but the opposite sign; EPC1 is chosen arbitrarily and used consistently in the rest of this study.

Over a large number of dwellings, the distribution of Y for a given value of <EPC> (shorthand $Y_{<EPC>}$) may be able to yield information about the measurement error defined in the previous section. This is because if measurement error were the only factor causing EPC1 and EPC2 to differ, then $Y_{<EPC>}$ would be expected to be distributed according to a Gaussian distribution with a standard deviation $\sigma_{<EPC>}$.

$$Y_{<EPC>} \sim N(0, \sigma_{<EPC>}) \tag{5}$$

The observed spread around one value of <EPC>, $\sigma_{<EPC>}$, could then be related to the measurement error, $\sigma$, providing that EPC1 and EPC2 are independent measurements:

$$\begin{aligned}
\text{var}(Y) &= \text{var}\left(\tfrac{\text{EPC1}-\text{EPC2}}{2}\right) \\
&= \tfrac{1}{4}\left(\text{var}(\text{EPC1}) + \text{var}(\text{EPC2})\right) \\
&= \tfrac{1}{4}\left(\sigma^2 + \sigma^2\right) \\
&= \tfrac{\sigma^2}{2} \\
\sigma_{<\text{EPC}>} = \text{s.d.}(Y) &= \tfrac{\sigma}{\sqrt{2}}
\end{aligned} \tag{6}$$

However, due to the aforementioned list of possible reasons why EPC2 may differ from EPC1 including genuine changes to the property, it is highly unlikely that measurement error is the sole reason for difference between EPC1 and EPC2 in all dwellings. Therefore, the task of the analysis is to explore in what regions of the data it is possible to estimate $Y_{<\text{EPC}>}$, the distribution around a given value of <EPC> whose variation is solely due to measurement error, distilling it from other causes of change between EPC2 and EPC1. Where $Y_{<\text{EPC}>}$ can be estimated, the measurement error can be predicted and the research question can be answered.

## 4. Methodology

The methodology to be used is semi-parametric statistical modelling [23]. This refers to an exploratory approach in which the researcher can be selective about where to apply models, based on the structure observed in the data. This is in contrast to the use of an off-the-shelf statistical model of variation, such as repeated measures ANOVA, which attempts to explain all variation within a dataset using a small set of parameters.

This approach is used for several reasons. Firstly, the fortunate situation of having a very large number of pairs of EPC ratings at most values of <EPC> means there is less need to make strong assumptions about relationships such as how error varies with different values of <EPC>. We are able to allow estimates of parameters relating to variation to depend on <EPC> in a very flexible way, by estimating these parameters separately for a given value of <EPC>. We do make use of a parametric assumption about the distribution of the error $\Delta$.

The second reason for this approach is that, as will be shown, there are asymmetries and other non-standard structures in the data which render certain assumptions in parametric statistics invalid; therefore, care must be taken before applying standard techniques across the whole domain of the data. Instead, the data can guide the course of the analysis and parametric assumptions can be introduced as appropriate.

The Analysis section is a step by step description of the analysis, in which the specific methods used in a given step are chosen after examining the results from the previous step.

## 5. Analysis

An overview of the analysis is as follows:

- The dataset of repeated measurements was visualised in terms of the variation in each dwelling's two EPC ratings around their mean, <EPC>. This revealed that the variation was not a consistent value but decreased with increasing building energy efficiency; it also highlighted areas in which Gaussian characterisation of error was invalid. (Section 5.1)
- Subsets of data were visualised at individual values of <EPC>, revealing the presence of two different distributions. These were hypothesised to arise from a combination of random error and energy efficiency upgrades. (Section 5.2)
- A decision was made as to how to distil the measurement error from the variation caused by energy efficiency upgrades. (Section 5.3)

This outline is expanded upon below.

### 5.1. Visualisation of Per-Dwelling Divergence in EPC Rating

As a first step towards applying the measurement error model in Section 3 to the data, it was necessary to further examine the per-dwelling variation in EPC ratings; specifically, to ascertain whether this is constant across the dataset, or whether the variation is itself variable.

The data was first visualised by plotting <EPC>-EPC1, the divergence of the measurements from their mean, against <EPC>. This is shown as a density plot in Figure 5.

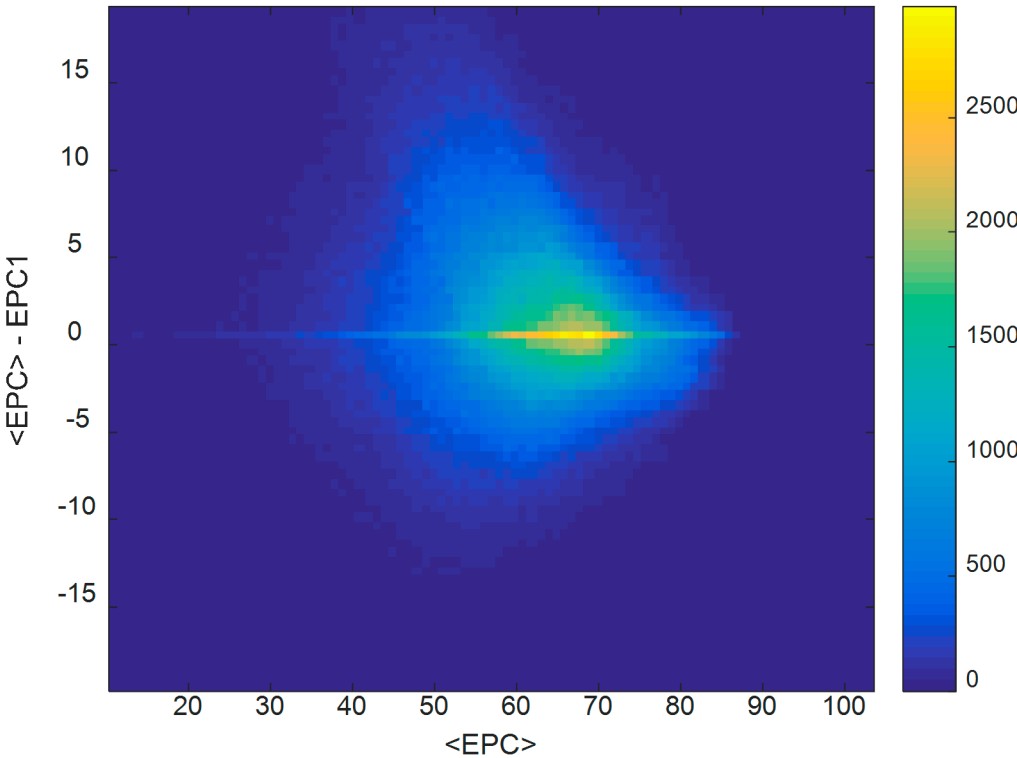

**Figure 5.** <EPC>-EPC1 plotted against <EPC>. Illustrating the divergence of per-house measurements from their mean.

There is a noticeable horizontal line along <EPC>-EPC1 = 0 on Figure 5, showing cases in which the two ratings per dwelling are identical. The cloud of points above and below the <EPC>-EPC1 = 0 line on Figure 5 provides an illustration of the spread of EPC measurements about their means. However, it is difficult to observe the extent of this spread at all values on the horizontal axis, due to most of the data being concentrated in a small region of the plot (<EPC> around 60–70), dominating the plot. Therefore, in Figure 6 the same data is shown as in Figure 5, but the colour scale normalises every point on the horizontal scale separately. Thus, each vertical strip of data (the spread of data at each value of <EPC>) is coloured separately, revealing the structure in the data in less-dense regions.

A number of observations can be made from Figure 6. At values of <EPC> between 40 and 85, there appear to be more vertical spread in the data at lower values of <EPC>. At values of <EPC> greater than around 85, there is an insufficient quantity of data to observe how it is distributed. It is therefore appropriate to discount this region as lacking a sufficient amount of data for analysis.

At values of <EPC> under 40, a boundary effect is present in the data, shown by the two diagonal lines. This arises from a maximum allowed divergence between two measurements for certain values of <EPC>, due to the lowest allowed EPC rating being 1. For example, a dwelling with <EPC> = 2 must have pairs of EPC1 and EPC2 that are equal to either 2 and 2, or 1 and 3, resulting in a maximum value of |<EPC>-EPC1| of 1.

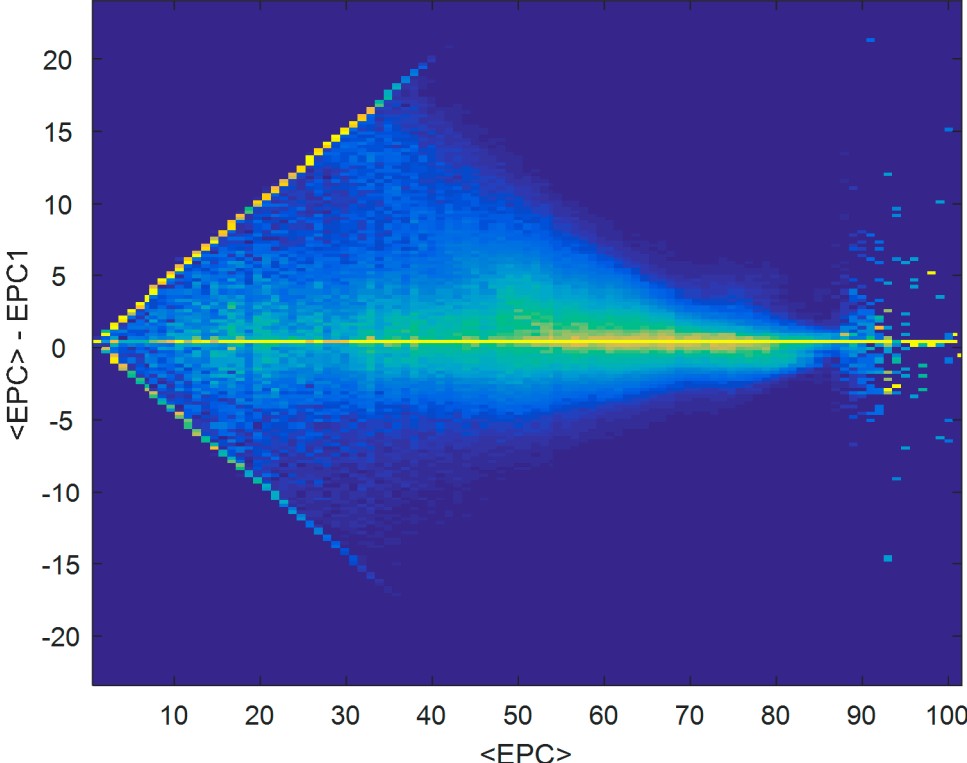

**Figure 6.** Normalised version of Figure 6, allowing the spread at every value of <EPC> to be observed. No colour bar is given, as in this plot a given colour does not map onto the same number of dwellings at all values on the *x*-axis.

Furthermore, according to the EPC documentation [24], if any rating is calculated as less than 1 EPC point (including negative ratings which are possible), the value is reported as 1. Figure 2 exhibits a spike in the distributions of both EP1 and EPC2 at 1; an explanation for this could be that thousands of dwellings had an original negative rating which was then reported as 1. This artificial upward correction creates a confounding effect, as illustrated in the following example. If the true value of an EPC is 2, and it is subject to a Gaussian error of ±10, typical measurements would be −8 and 12. Since any measurements below 1 are moved upwards, the measurements would get recorded as 1 and 12, giving a mean of 6 or 7 depending on rounding. Therefore, if the error is larger than the true value, then <EPC> does not give a good estimate of the true value. In this region of the data, the correction process applied to the data invalidates the Gaussian assumption of error.

It is common in statistical analysis to address problems introduced by bounded scales by transforming the data such that it lies on an unbounded scale, e.g., via a logit transformation. However, the EPC dataset is not a typical example of a dataset lying on a bounded scale, since its bounding is artificial—some data points have been increased from potentially negative values up to 1. Mathematical transformation of the analysis scale therefore does not properly replicate this data manipulation process. Furthermore, transformation of the analysis scale would render the interpretability of the parameters opaque with respect to the theory set out in Section 3. Therefore, the analysis will continue on the original scale, but the interpretation of results will not be carried out at low values of <EPC> where the Gaussian model of error presented in Section 3 is not valid.

Despite the behaviour at the extreme low and high ends of the <EPC> scale in Figure 6, it was observed that vertical spread appears to decrease with increasing <EPC>. The next step further investigates this trend.

### 5.2. Examination of the Spread of Data around Each Mean EPC Value

In Figure 7, subsets of data from different values of <EPC> in Figure 5 are presented. These are effectively vertical slices through Figure 5: this time, the *x*-axis is <EPC>-EPC1, showing the dispersion of data around each dwelling's mean. The red lines represent the boundaries beyond which <EPC>-EPC1 cannot fall.

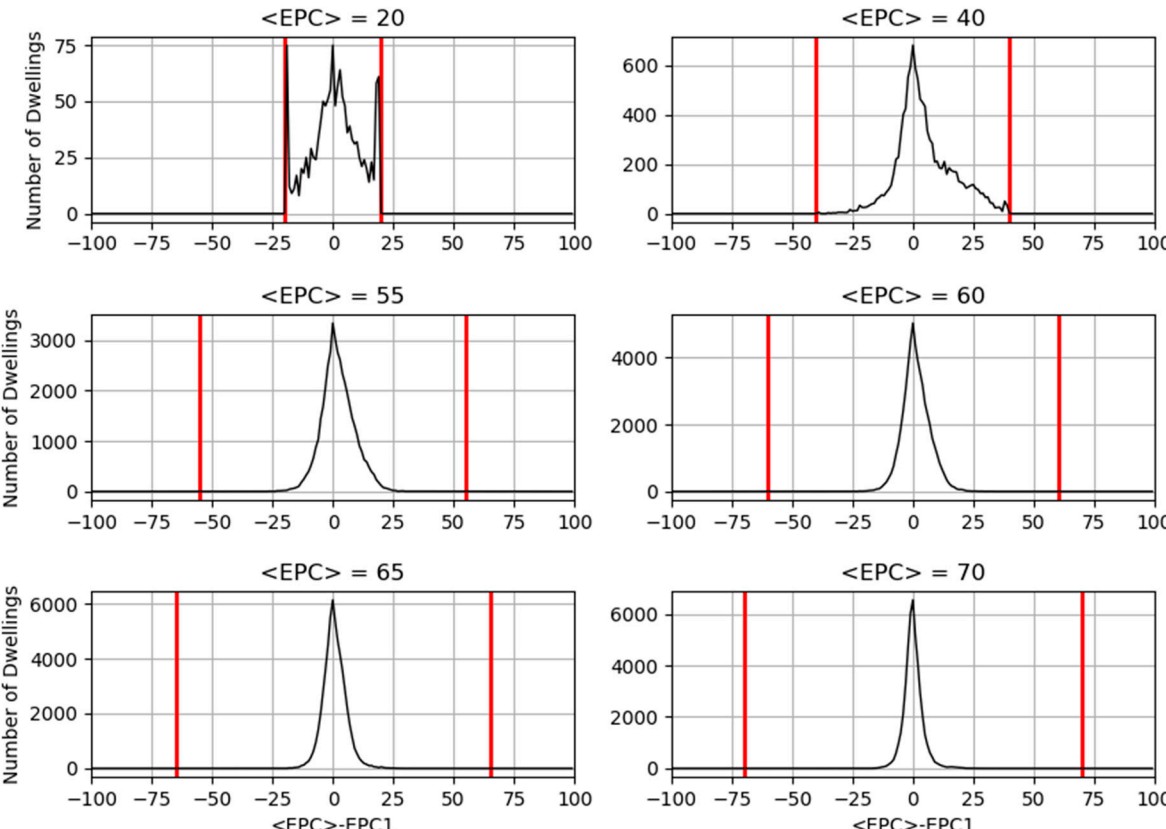

**Figure 7.** Spread of pairs of measurements around their mean, using subsets of data with a common <EPC>. Red vertical lines show the boundaries within which <EPC>-EPC1 must lie due to the scale boundaries.

Three features of the data are evident from Figure 7. Firstly, the difference between subplots shows that the distribution of <EPC>-EPC1 narrows for greater values of <EPC>. Secondly, it reveals that for a given <EPC>, the distribution of <EPC>-EPC1 deviates from a Gaussian form in that its peak is sharp instead of smooth, as was observed in Figures 5 and 6. Thirdly, the distribution of <EPC>-EPC1 above zero has a wider distribution and a longer tail than that below zero.

To further explore the third observation, the data in subplot in Figure 7 was fitted with two separate half-Gaussian distributions with scale parameters $\sigma_L$ and $\sigma_R$, on the left and the right of zero respectively. $\sigma_L$ is the scale parameter for the distribution of dwellings whose EPC rating decreased on their second assessment; $\sigma_R$ represents those whose rating increased. This is shown in Figure 8.

Figure 8 shows that $\sigma_L$ and $\sigma_R$ describe the data well, except for the following cases:

- The top-left subplot, in which <EPC> = 20, has extra peaks at −19 and 19. This is another illustration of the boundary effect observed in Figure 6, due to a high number of measurements reported at 1 as seen in Figure 2. However, when <EPC> increases to 40 as in the top-right subplot, this effect becomes negligible.
- Sharp peaks are present in each subplot at <EPC>-EPC1 = 0. This points to an excess of cases in which EPC1 = EPC2 compared to that expected, which is likely explained by duplicate certificates.

Obtaining the values of $\sigma_L$ and $\sigma_R$ in Figure 8 demonstrates that the distribution of dwellings whose rating decreases on second assessment is narrower than the distribution of dwellings whose rating increases. This effect is less evident as <EPC> increases; at <EPC> = 70, the left and right hand sides of Figure 8 are approximately symmetrical.

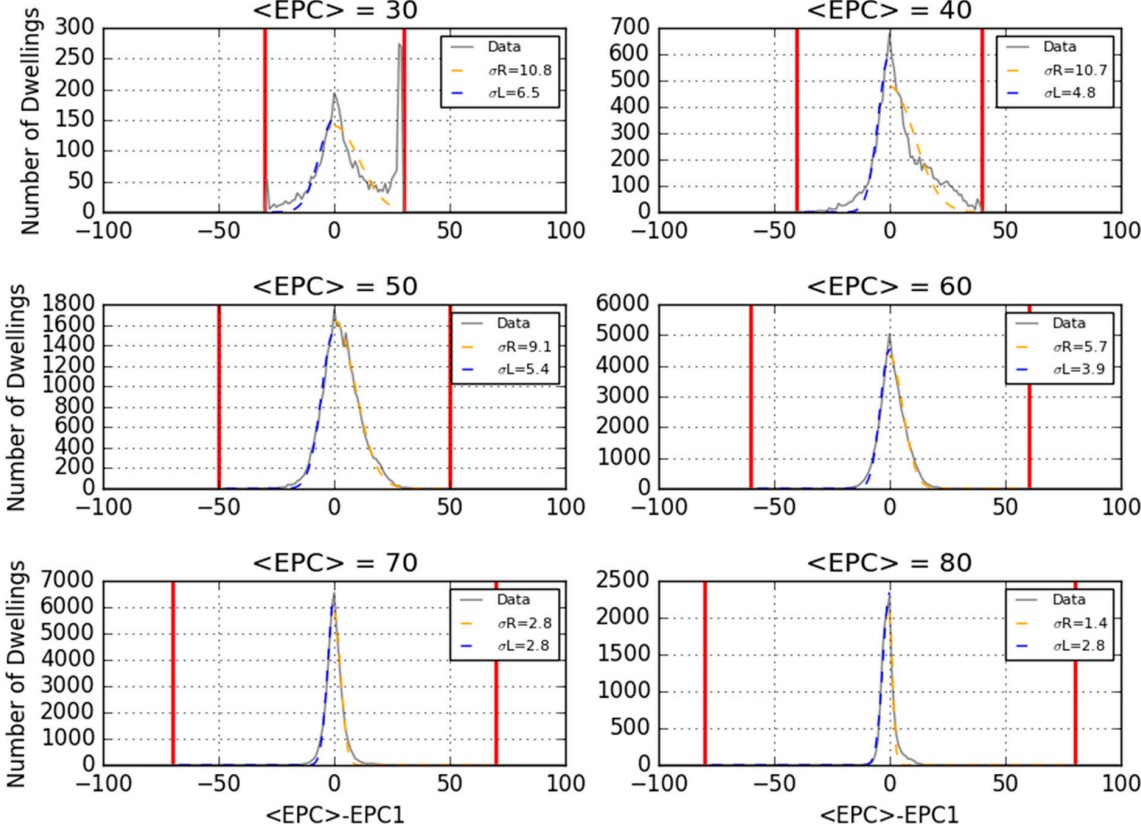

**Figure 8.** Distribution of <EPC>-EPC1 for certain values of <EPC>. Data fitted with a different half-Gaussian distribution to the left of zero ($\sigma_L$) and right of zero ($\sigma_R$).

The differing values for $\sigma_L$ and $\sigma_R$ within each subplot in Figure 8 have important implications. Since the measurement error was theorised in Section 3 to manifest in this dataset as a symmetric distribution of residuals around any given value of <EPC>, Figure 8 shows that error alone does not explain all of the variation around <EPC> observed in the data. Additional effects are present, causing more EPCs to increase than decrease upon second measurement. A plausible explanation for the additional effects is the impact of energy efficiency retrofit measures which will have been carried out on some properties.

### 5.3. Decision of How to Characterise Measurement Error

In order to characterise the measurement error, its effect must be separated out from that of energy efficiency upgrades. It is not possible to reliably identify upgrades on an individual dwelling basis since the available information in the dataset on building fabric, heating system etc. is subject to the same type of error as the EPC ratings. However, since upgrades are very unlikely to result in a reduction in EPC rating, it was assumed that where EPC ratings decrease upon second measurement, the difference between EPC1 and EPC2 is due to measurement error alone, whereas EPC ratings increasing upon second measurement result from a combination of measurement error and upgrades. According to this assumption, $\sigma_L$ characterises the spread due to measurement error, whereas $\sigma_R$ arises from a combination of measurement error and building upgrades.

Having specified $\sigma_L$ to be the part of the observed variation due to measurement error, $\sigma_L$ can then be used in Equation (5) as the term $\sigma_{<EPC>}$. This equation states that the measurement error on a rating, $\sigma$, is simply related to $\sigma_{<EPC>}$ by a factor of $\sqrt{2}$, so we can now calculate the measurement error for a given EPC rating.

### 5.4. Applying the Error Model

Finally, $\sigma_L$ was estimated separately for every value of <EPC>, without attempting to parameterise how $\sigma_L$ varies with <EPC>. It was then decided to retain the results only for EPC ratings between 35 and 85. This is because, as demonstrated in Figures 6–8, the error model is not valid in regions where the EPC scale boundaries interfere with the model's Gaussian assumptions (particularly affecting the bottom end of the scale), or where there is too little data (particularly affecting the top end of the scale).

In summary, the parametric assumption that measurement error results in a Gaussian distribution of data around a given value of <EPC>, and the lack of use of a parametric assumption for how this error might vary with <EPC>, combine to complete the semi-parametric characterisation of measurement error. The results are presented in the next section.

## 6. Results

### 6.1. Measurement Error

The key result of this study is the estimated measurement error on EPC ratings. This is shown in Figure 9 for the part of the EPC scale in which it was deemed valid to apply the error model (as described above), i.e., between 35 and 85. It can be seen that the error generally decreases with EPC rating. This decrease is not smooth and even reverses sign in some areas, but the general trend downwards is clearly observed.

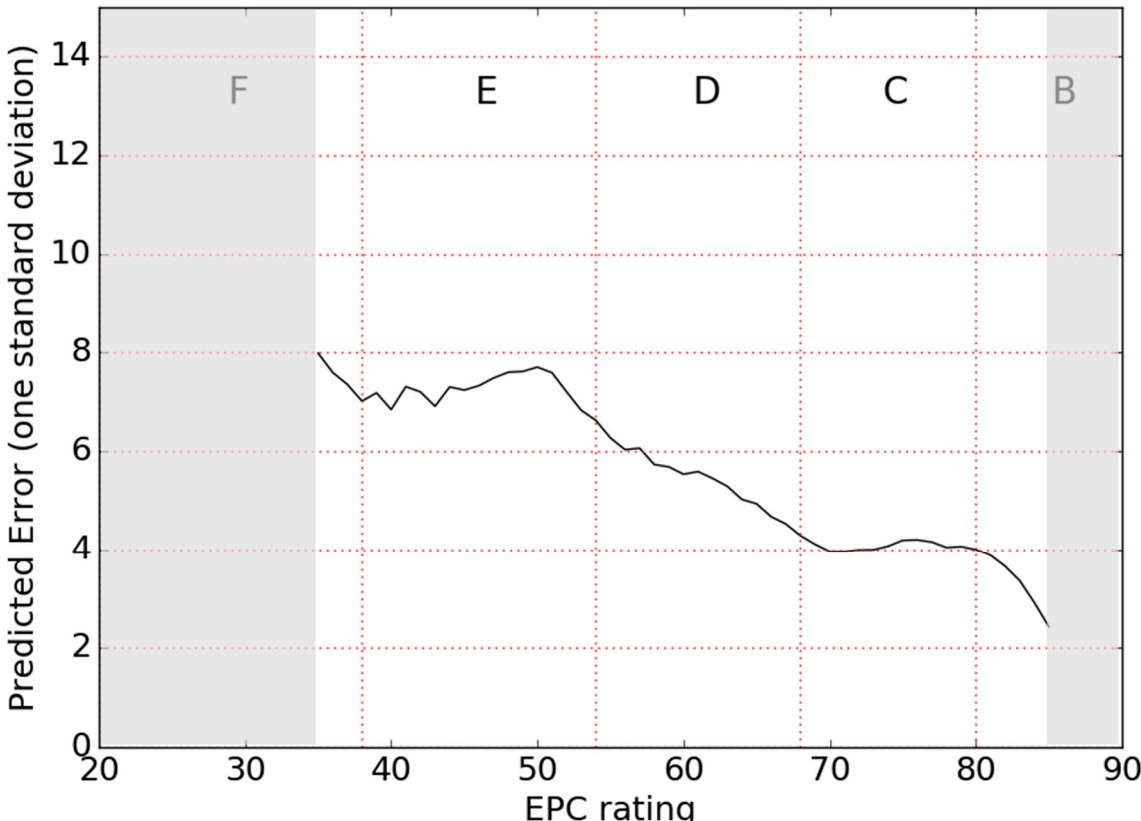

**Figure 9.** Predicted one standard deviation error for different EPC ratings. Grey areas are those in which assumptions are not valid, and therefore, the error is not estimated.

Key values from Figure 9 are summarised in Table 2, reported to one and two standard deviations respectively.

**Table 2.** Predicted error on certain EPC ratings. One and two standard deviation values given.

| Rating | 38 | 46 | 54 | 61 | 68 | 74 | 80 | 85 |
|---|---|---|---|---|---|---|---|---|
| Band | E/F Boundary | Middle of E Band | E/D Boundary | Middle of D Band | D/C Boundary | Middle of C Band | C/B Boundary | Middle of B Band |
| Error (1 s.d.) | 7.0 | 7.3 | 6.6 | 5.6 | 4.3 | 4.1 | 4.0 | 2.4 |
| Error (2 s.d.) | 14.0 | 14.7 | 13.2 | 11.2 | 8.6 | 8.1 | 8.0 | 4.8 |

The interpretation of the error values in Table 2 is illustrated using an example as follows. For a dwelling rated by an assessor at 61 EPC points, in 68% of cases the true value would lie within 5.6 EPC points of this rating (55.4, 66.6), and in 96% of cases within 11.2 EPC points (49.8, 72.2).

The error model is not valid for ratings in the G band and most of the F band, since the spread of data around the mean rating per dwelling in these regions was found to be so large that it was truncated by the lower bound of 1. The fact that this significantly affected all values of <EPC> under 35, not just values close to 1, indicates that very large errors may be present on ratings in bands F and G.

*6.2. Impact of Estimated Measurement Error on EPC Band*

The error estimated in this analysis may be large enough to affect which EPC band a dwelling is placed in. The potential impact of this effect is explored in this section.

The error associated with each EPC rating in Figure 9 can be used to calculate the probability that a dwelling with a given 'true value' is rated in the next band up, by calculating the area under the Gaussian distribution around the rating which spreads out into the next band. For example, a dwelling whose true value is 46 (in the middle of the E band) can be shown to have a 12% chance of being rated in the next band up, i.e. the D band.

Whilst it would be useful to estimate the proportion of dwellings which have been placed in a higher band than they should be, due to error alone, this is not possible within this analysis framework, since the error model cannot be applied to all EPC ratings. However, some examples for the region where this analysis is valid are discussed below. The probability that dwellings with their true values in one band are placed in a different band when assessed can be calculated using the sets of ratings and errors in Figure 9 weighted by the distribution of dwellings within the 'true band'. The distribution of true values of EPC ratings is unknown, so the distribution of mean rating per dwelling (distribution of <EPC>) is used as a proxy for weighting purposes. Results are shown in Table 3.

**Table 3.** Estimated chance of misclassification of band.

| | | |
|---|---|---|
| **Misclassification to a better band** | Probability that true 'E' dwellings are rated at D or above: | 24% |
| | Probability that true 'D' dwellings are rated at C or above: | 15% |
| **Misclassification to a worse band** | Probability that true 'C' dwellings are rated at D or below: | 19% |
| | Probability that true 'D' dwellings are rated at E or below: | 15% |
| | Probability that true 'E' dwellings are rated at F or below: | 13% |

Two effects are occurring simultaneously in Table 3. Firstly, as error decreases with increasing true EPC rating, the band misclassification probability decreases, as seen in the first two lines of the table. However, the distribution of ratings in general centres around band D, meaning that for example there are fewer dwellings in the bottom of band E than the top, and more dwellings at the bottom of band C than the top. Thus, if a dwelling's true rating falls within band F, it is more likely to be 54

than 39. This weighting causes the trend that the misclassification probability in the bottom three lines of Table 3 decreases.

### 6.3. Energy Efficiency Upgrades

After accounting for the spread in the data attributed to measurement error, the remaining spread in per-dwelling EPC rating was examined. The residual of the data minus the semiparametric error model is shown in Figure 10, highlighting two main regions in which the simple model of error does not describe the data well. These regions are discussed in this section.

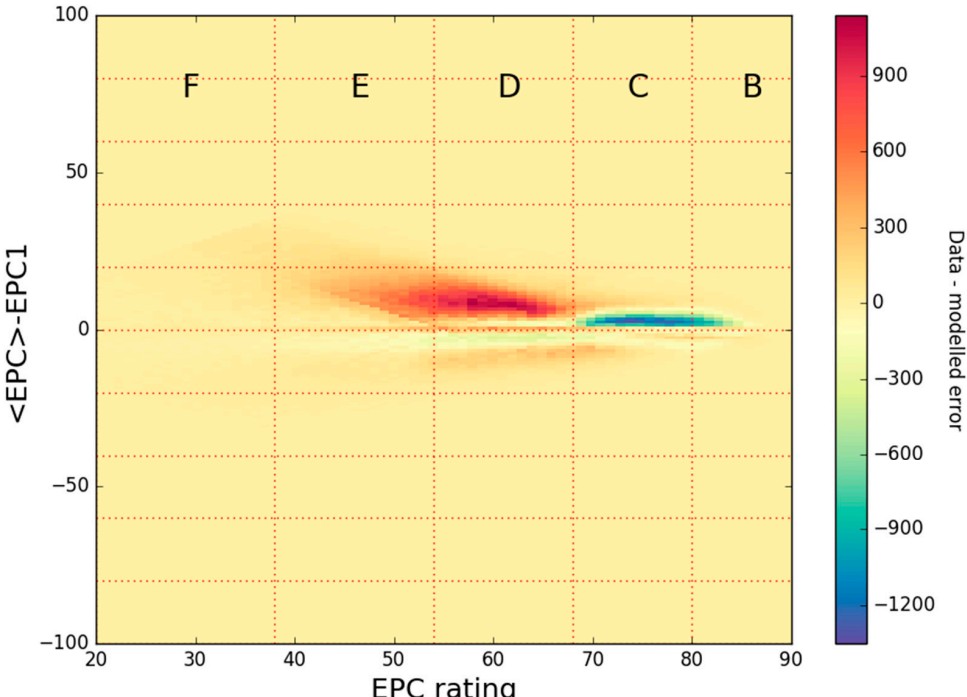

**Figure 10.** Data minus modelled error, highlighting areas in which the semiparametric error model does not predict the data well. Regions in which there is more data than the error model predicts are shown in red, regions in which there is less data than the model predicts are shown in blue.

The first area of discrepancy between the data and error model is the red area in the E and D bands in Figure 10, representing more upward movement of EPCs upon second measurement than would be expected from error alone. This feature of the data is further illustrated in Figure 11, which shows the second EPCs for all homes where the first EPC is equal to 40. The distribution has a step change at the D/E band boundary, suggesting the likely presence of a target of D for the upgrade of the properties. Step changes do not normally arise naturally in distributions of measurements—achieving exactly the required value in this way would require the assessor's involvement beyond simply measuring and inputting data. This is reflected upon in the Discussion.

The second main area where the error model does not describe the data well is the blue area in Figure 10. In this region, the first EPC was relatively high, but more properties than expected by the error model received a lower second EPC rating. The reason for this is unknown, but may be associated with changes to the SAP model, revisions to the expected performance of specific dwelling features within model inputs, or to changes in the method of assessment.

Further investigation of the nature and extent of energy efficiency upgrades is not pursued in this article. This is primarily because the relevant data—the descriptions of building elements, heating systems, etc.—is affected by measurement error, so it cannot immediately be used as a true description of the state of the building. However, further work could potentially apply the same method as used

in this study to the input data, in order to attempt to statistically separate random variation in inputs from systematic improvements.

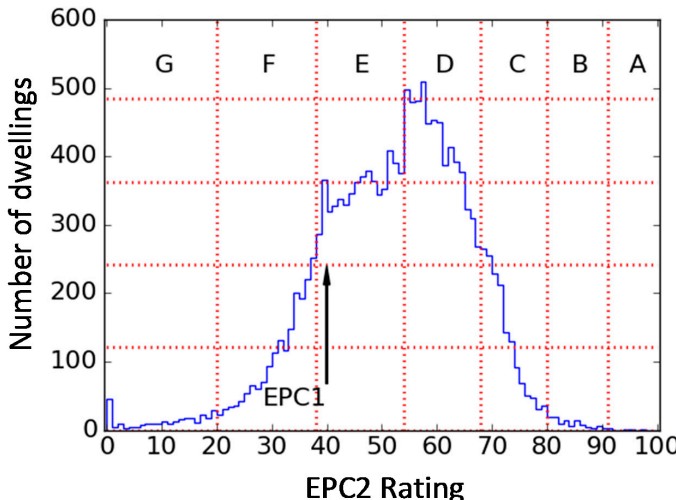

**Figure 11.** Second EPCs for dwellings with a first EPC of 40. Distribution shows a spike at exactly 40 and a step change at the D/E band boundary.

*6.4. Validity of Findings*

The research objective was to derive a value for measurement error consistent with the observed data. This has been achieved for EPCs and been shown to generally decrease with increasing EPC rating.

The research question was phrased to ensure it was answerable using a mathematical approach. However, the external validity of the analysis is a much larger challenge. It is not possible to prove that the simple premise underlying the analysis is correct: that the measurement error can be characterised using those EPC ratings which decrease upon second measurement. This is likely to be the case, due to prior knowledge of the existence of this error in the dataset [17], but the non-smooth nature of the relationship between error and EPC rating in Figure 9 may indicate that additional mechanisms beyond measurement error have also been encapsulated in the analysis.

## 7. Discussion

This research set out to quantify the uncertainty on the process of generating EPC ratings, here termed the measurement error. This uncertainty was found to be not one single value but to generally decrease with increasing building energy efficiency. Across the part of the dataset, where the error model could be applied, the one standard deviation error varied from 8 EPC points at the upper end of the F band to 2.4 in the B band. The error in the F and G bands was so high that the error model could not be applied, therefore, although the value cannot be quantified within this analysis framework, we can signal the existence of a problem.

These results are now contextualised. Firstly, the estimated error is compared with UK guidelines. Government guidance requires that 95% of a sample of assessments yield EPC ratings within ±5 EPC points of the 'truth' [15]; this corresponds to just under two standard deviations. In contrast with previous work [18], this analysis shows that error is generally larger than the recommended value except where buildings are at or higher than the middle of band B. Note that this conclusion does not apply to individual assessors but to the average assessment.

Secondly, the implication of the error for policy making is discussed. In the UK, the EPC band is often used as a basis for individual and stock level energy efficiency and fuel poverty policies. For example, the UK government's Clean Growth Strategy uses EPC banding as the metric of stock performance, aspiring to the aims of "*all fuel poor homes [ . . . ] upgraded to Energy Performance Certificate (EPC) Band C by 2030*" and "*as many homes as possible to be EPC Band C by 2035 where practical, cost-effective*

*and affordable.*" [25] A second policy example of the use of EPC bands is the recent regulation forbidding properties in the lowest bands (G and F) from being let out [7]. This analysis predicted that 24% of E dwellings may achieve a D by chance, and 15% of D dwellings may achieve a C by chance, highlighting the susceptibility of the band policy framing to the misidentification of properties.

Thirdly, the implications outside the UK are considered. Most EU member states use a method of producing EPCs involving building assessments [11]. To the authors' knowledge, there is currently no equivalent quantification of uncertainty for another country, therefore, the magnitude of the error predicted here provides insight into the potential error in regions with similar methodologies. Any such comparison must be undertaken with caution since each country's specific method within the overall calculation methodology is different. It is also notable that the cost of an English/Welsh EPC is towards the lower end of the European range [11], which, if reflected in the rigour of the assessment, may impact on the accuracy of the resulting EPC compared to those in other countries.

Fourthly, the predicted error from the current assessment process is an important measure by which to assess the performance of alternative, empirical rating metrics. There is a growing interest in the use of empirical data to construct energy ratings [26], partly due to the uptake of smart meters in Europe which will enable higher resolution data collection than has previously been possible on a large scale [27]. Unlike calculated methods, empirical rating methods are accompanied by a set of challenges related to standardisation, and therefore, the characterisation of uncertainty is equally important in these new methods. Whilst it is hoped that they will outperform the calculated method, the error predictions in this study provide a reliability baseline which future energy efficiency rating methods should exceed.

We now discuss the evidence of refurbishment measures being undertaken between subsequent EPC ratings of properties in the dataset. Subtracting the estimated effect of measurement error from the data left an asymmetric structure likely to represent the energy efficiency upgrades carried out between subsequent EPC assessments. Although the presence of error renders dwelling-level insights difficult to ascertain, features are apparent across the whole dataset. One is the existence of a systematic effect causing dwellings with low first EPCs to attain a D or C band on their second measurement, observed through a step change in the distribution of second EPCs at the D/E boundary. The large number of dwellings attaining a D or C on second measurement is possibly explained by retrofit interventions on these properties, however, the observed step change is an unusual feature of genuine data distributions [28] and may indicate some manipulation. Discontinuities from one band to another in post-retrofit EPC results have been observed before: Collins et al. [29] documented a 'bunching' discontinuity effect in Irish EPC data from a sample of dwellings having undergone grant-funded retrofit works. This 'spike' form of discontinuity differed from the step change observed in the England and Wales dataset; nevertheless, we agree with the statement by Collins et al. highlighting the importance of understanding whether the effect '*is real and reflects accurate assessments of properties' energy performance or the result of illicit activity*'. More research is required to investigate this issue.

This study, quantifying total uncertainty, was a first step to a better understanding of the error on EPCs; further work would investigate the presence, importance and causes of different sources of error. For example, Hardy et al. identified flats and maisonettes as being frequently associated with built form error [18] and offered a potential explanation that the SAP methodology does not use the built form of flats, leading to assessors paying less attention to this. In the current study, inefficient buildings were found to have higher levels of error; this is possibly because of error propagation through multiplying variables together in the SAP calculation. For example, in a thermally inefficient building, a heating system efficiency which is incorrectly input would change the predicted energy use a high absolute amount compared to the effect of the same mistake in a thermally efficient building.

## 8. Conclusions

EPC ratings and their associated predictions of energy consumption are used throughout the UK and EU for a range of policy, landlord and homeowner decisions. This study used a set of 1.6 million

dwellings with two EPC certificates to produce a national-level estimate of uncertainty arising from the assessment and calculation process of generating EPC ratings for existing dwellings in England and Wales. This was carried out by applying a simple model of error to the subset of the data with two EPCs per dwelling. The data was explored to determine how to apply this model; the chosen implementation was a semiparametric approach which allowed error to vary with EPC rating in regions of the data where the model assumptions were valid.

The predicted error (one standard deviation) was found to generally decrease with EPC rating, from 8 EPC points at the upper end of the F band to 2.4 in the B band. This error leads to a significant probability of mis-classification of dwelling band and its associated policy consequences, for example, a 13% probability of an E dwelling being rated as F or lower, and therefore, being unable to be rented out.

The analysis also highlighted features of the differences between subsequent EPC assessments not captured by the model of measurement error, notably a transition from low first EPCs to second EPCs in the D and C bands. This is likely to be caused by retrofit interventions occurring between the assessments; however, the uncertainty on the assessment data means that identifying interventions is not straightforward. Further work would focus on the input data, attempting to isolate retrofit interventions from both measurement error and the step change in EPC ratings at the E/D boundary, which could be driven by the presence of a target to achieve band D.

This is the first study to estimate at a national scale the total uncertainty on EPCs in England and Wales. The level of imprecision found challenges their use in supporting minimum standards for rental properties and energy efficiency retrofit measures, and in indicating the performance of dwellings to potential tenants and buyers. This result could also be relevant for other countries which also use the 'calculated' method of producing EPC ratings.

The findings of this study suggest that revision to such energy performance assessment methods may be necessary; this may either be achieved by improving the current assessment and modelling method, or developing assessment methods based on actual energy data. The development of reliable methods to assess the energy performance of buildings is challenging, with each alternative technique facing its own limitations and issues. However, a method which can be demonstrated to produce reliable EPCs across a national building stock, performed at an acceptable price, can underpin both policy and homeowner decision making, to support decarbonisation of the energy system.

**Author Contributions:** Conceptualization and main author: J.C., methodology: P.J.N. and J.C., software and analysis: P.B., editing: C.E., T.O. and J.W.

**Funding:** This work was funded by the Engineering and Physical Sciences Research Council (EPSRC) funded RCUK Centre for Energy Epidemiology under EP/K011839/1 and UKRI Centre for Research in Energy Demand Solutions under EP/R035288/1.

**Conflicts of Interest:** The authors declare no conflict of interest.

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
