# Peer review of "Quantifying the Measurement Error on England and Wales EPC Ratings"

_energies, doi:10.3390/en12183523_

Round 1

Reviewer 1 Report

Considering the limitation of repeated rating on a single dwelling, the approach by the authors to estimate the measurement error of EPC rating is practical.

In reality, dwellings belong to one EPC band should have certain close physical conditions, so each band should has its own error range.   In low score region, intuitively there could be large spreading error, so an overall decreasing error trend shown in Fig.9 is convincing.  Beyond the 85 score, although with much less data, the authors could employ this decreasing trend and give an estimation of the error towards the highest score region.

The overall structure of the paper is good and well written.

Reviewer 2 Report

The paper presents a novel error analysis that can estimate the size of the measurement error, identified in the process of creating an Energy Performance Certificate. The paper focuses on the assessments of 1.6 million existing dwellings in England and Wales. A statistical model of how 15 measurement error contributes to variation between repeated measurements was set out, and 16 exploratory data analysis was used to decide how to apply this model to the available data.

I consider the paper to be very valuable due to increased concerns in many EU member states regarding the way that the Energy Performance Certificates are elaborated for new or existing buildings, considering that some of them underperform compared to the values issued in the EPC.

Although that the authors have a good start in assessing the errors that show invalid results in the EPC, the fact that the energy performance assessment methodology varies across Europe, the values or maybe even the methodology recommended in the paper could be very difficult to apply in other countries. 

Another observation is that a second EPC, after applying retrofit measures, should always give better results for a building. Only in the case of existing building, were no retrofit measures were applied, but a second EPC was issued, can have worse results (i.e. a worst band) compared to the first certificate. Thus, from my point of view, this two sets of buildings should be set apart.

Some minor changes that I want to recommend is to verify the paper template and some typos or doubled words.

Some important improvements can be:

The assumption "If no such changes have been made then the second EPC rating should be equal to the first. Therefore, EPC rating should not decrease over time." Such assumptions are not correct considering that the building is exposed to boundary conditions that effect the thermal and mechanical properties in time. Thus, a EPC emitted in 2019 can not have the same value as the one emitted in 2023. This is what we find on the market. For the reader, Figure 3 is difficult to understand. Why at each column are one of the letters (segments) highlighted?
Please either add some helping signs to understand the changes form the first EPC to the second, or simplify the chart. Also, please add more explanations. What about the number at the tops. For example 627 corresponds to the buildings in band G? How can the reader identify which is the original band? By "EPCtrue is the theoretical true value which would be obtained with a perfect assessor" you refer to the result obtained for the notional building (i.e. reference building)?  How were the percentages calculated " 68% of cases the true value would lie within 5.6 EPC points of this rating (55.4, 66.6), and in 96% of cases within 11.2 EPC points (49.8, 72.2)". The same applies for table 3 In the sentence "Empirical rating methods are accompanied by a set of challenges related to standardisation (which calculated methods avoid) ..." it looks like a word is missing
